# Use of Monoclonal Antibodies in Pregnant Women Infected by COVID-19: A Case Series

**DOI:** 10.3390/microorganisms11081953

**Published:** 2023-07-31

**Authors:** Pietro Crispino, Raffaella Marocco, Daniela Di Trento, Gloria Guarisco, Blerta Kertusha, Anna Carraro, Sara Corazza, Cristina Pane, Luciano Di Troia, Cosimo del Borgo, Miriam Lichtner

**Affiliations:** 1Medicine, Santa Maria Goretti Hospital, Via Scaravelli Snc, 04100 Latina, Italy; 2Unit of Infectious Disease, Sapienza University of Rome, 04100 Latina, Italy; rmarocco@libero.it (R.M.); ddtrrento@libero.it (D.D.T.); kkertusha@libero.it (B.K.); acarraro@libero.it (A.C.); scorazza@libero.it (S.C.); cdelborgo@libero.it (C.d.B.);; 3Unit of Diabetology, Santa Maria Goretti Hospital, 04100 Latina, Italy; gguarisco@libero.it; 4Unit of Gynecology, Santa Maria Goretti Hospital, 04100 Latina, Italy; cpane@libero.it (C.P.); lditroa@libero.it (L.D.T.)

**Keywords:** COVID-19, pregnancy, monoclonal antibody (Mabs)

## Abstract

Background: Monoclonal antibodies are designed to target specific proteins of COVID-19 and can be used as a treatment for people with mild to moderate infection and at a high risk of severe disease. Casirivimab/imdevimab, sotrovimab, and Bamlanivimab/etesevimab have been authorized for emergency use in the treatment of COVID-19. However, during pregnancy, these drugs have not been extensively studied. Methods: A total of 22 pregnant women with mild to moderate infection were treated with three different monoclonal antibodies, and efficacy and safety were evaluated in the first period and until six months of follow-up. Results: No infusion/allergic reactions occurred. No fatal or adverse events were observed in the pregnant women or fetus. The time of negativization with sotrovimab was shorter in comparison to Imdevimav/casirivimab (*p* = 0.0187) and Bamlanivimab/etesevimab (*p* < 0.00001). The time of negativization with sotrovimab was earlier in comparison to Imdevimav/casirivimab (t-value: 2.92; *p* = 0.0052) in vaccinated patients and similar in comparison to Imdevimav/casirivimab (t-value: 1.48; *p* = 0.08). In unvaccinated patients, sotrovimab was faster to achieve negativization in comparison to Bamlanivimab/etesevimab (t-value: 10.75; *p* < 0.0005). Conclusions: Pregnant COVID-19 patients receiving sotrovimab obtained better clinical outcomes. Pregnancy or neonatal complications were not observed after monoclonal treatment, confirming the safety and tolerability of these drugs in pregnant women.

## 1. Introduction

A few years after the beginning of the pandemic and shortly after its formal conclusion, official data indicate that the prevalence of COVID-19 infection during pregnancy is comparable to that of the general population and that vertical transmission of the viral infection from the mother and fetus is considered a rare event [1,2], also linked, in general, to alterations in the early maternal and neonatal oral microbiome [3]. However, for preventive purposes, pregnant women have been included among the risk categories for major consequences of a related COVID-19 infection [1,2,3,4,5,6]. Furthermore, pregnant women have been excluded from clinical pharmaceutical trials of new drugs to be used against the virus, resulting in poor documentation of the complications and consequences of infection during pregnancy [2,3,4,5,6,7,8,9,10,11,12,13]. Nonetheless, the results of some studies showed that COVID-19 infection during pregnancy was associated with a greater probability of maternal, fetal, and neonatal adverse events [14,15,16], as more cases of preeclampsia, hospitalization in intensive care, preterm birth, and neonatal mortality were recorded compared to pregnant women not affected by the virus [17]. Furthermore, in two meta-analyses, newborns of infected mothers had a higher risk of hospitalization in the neonatal intensive care unit, with a lengthening of the mean duration of hospitalization compared to those born to uninfected mothers [14,16]. In particular, women older than 25 years, with metabolic syndrome, preconception obesity, or with the presence of chronic lung disease or arterial hypertension, were at greater risk of complications from a related COVID-19 infection than women without such conditions [18]. Considering that pregnant women undergo physiological changes that make them more susceptible to viral infections and are often limited in taking drugs [19], even in the face of severe manifestations of COVID-19 infection [20], and furthermore, considering the decrease in response to vaccines induced by continuous viral mutations [21,22], it was necessary to introduce new therapies capable of directly countering COVID-19 and thus prolonging the immunogenic effect of vaccines [5]. Among these drugs, combinations of injectable monoclonal antibodies such as Imdevimab/casirivimab, Bamlanivimab/etesevimab have been introduced for the treatment of COVID-19 in the most at-risk categories of patients [23]. Recently, sotrovimab, an engineered human monoclonal antibody, by recognizing a highly conserved and specific epitope of the spike protein, has demonstrated high antiviral potency and a greater induction of immune-mediated viral clearance in vitro [24,25]. Taking into account the therapeutic potential of each of these drugs, the main aim of our study was to describe the efficacy and safety of monoclonal agents in the treatment of COVID-19 infection in a series of pregnant women.

## 2. Patients E Methods

### 2.1. Study Design

We retrospectively evaluated a group of pregnant patients in the second and third trimester of gestation who consecutively came to our center to undergo early treatment for mild to moderate degrees of COVID-19 infection. Therefore, this was a single-center observational study conducted in accordance with the principles of the Declaration of Helsinki and the international ethical guidelines of the Council for International Organizations of Medical Sciences. All patients or their representatives provided written informed consent and a database with archived data is included on the basis of institutional forms required by the Italian health authorities. The study was approved by the local ethical committee (Comitato Etico Lazio 2, Approval code: ID Prot. 19/2022, Approval date: 18 January 2022). The risk factors considered for the inclusion of patients in the study and who were therefore eligible for monoclonal antibody therapy included: body mass index (body mass index, BMI) ≥30, or >95th percentile by age and gender; chronic renal failure, including peritoneal dialysis or hemodialysis; uncontrolled diabetes mellitus (HbA1c > 9.0% 75 mmol/L) or with chronic complications; primary or secondary immunodeficiency (e.g., neoplastic disease, leukemia or lymphoma, myeloma, autoimmune pathologies, HIV+/AIDS, malnutrition, pharmacological therapies, radiotherapy/chemotherapy, medicines immunosuppressive); cardio-cerebrovascular disease (including hypertension with concomitant organ damage); chronic obstructive pulmonary disease and/or other chronic respiratory diseases (e.g., individuals with asthma, fibrosis pulmonary or requiring oxygen therapy for reasons other than SARS-CoV-2); chronic liver disease (with the following warning: “monoclonal antibodies have not been studied in patients with moderate or severe hepatic impairment”); pathologies of neurodevelopment and neurodegenerative pathologies; and finally, hemoglobinopathies (Table 1). Patients with mild to moderate symptoms of recent onset (within 72 h and no later than 7 days) were considered eligible. Failure to adhere to the vaccination campaign was not a reason for exclusion from monoclonal antibody therapy.

### 2.2. Patients

Patients included in this observational study were older than 18 years and tested positive for SARS-CoV-2 by the reverse transcriptase-polymerase chain reaction (RT-PCR) test or an antigen test with a period of symptom onset not exceeding the previous 5 days. The study population was represented by pregnant women in their second and third trimester gestational periods and therefore represented patients at a high risk of progression or complication of COVID-19 infection. Therefore, patients with severe risk factors such as severe decompensated diabetes, obesity (body mass index > 30), severe chronic kidney disease (estimated glomerular filtration rate < 40 mL/min/1.73 m^2^), signs of heart failure congestive (≥New York Heart Association class II), and severe chronic lung disease were excluded from the study. Furthermore, any cases of hospitalization attributable to signs or symptoms of severe COVID-19 (dyspnea at rest, oxygen saturation level < 92%, or need for oxygen enrichment with supplemental administration) were excluded from the analysis (Figure 1).

### 2.3. Therapeutic Intervention

The verification of the suitability of the available therapeutic protocols and the treatment modalities themselves (type of drug, daily dose, number of daily administrations, and treatment time) were decided within 24 h before entering the study and the first administration of the drug. Using an Excel data-collection system, we retrospectively evaluated all pregnant patients treated in our center, considering the availability of pharmaceutical preparations:

(a) Imdevimab 1200 mg/casirivimab 1200 mg or Bamlanivimab/etesevimab 700 mg/700 mg;

(b) Sotrovimab 500 mg/8 mL;

Patients were observed for approximately 2 h after the infusion. Patients were reviewed 1 week and 1 month after monoclonal antibody administration, at delivery, and 6 months after delivery (Figure 1).

### 2.4. Aims Identification

For the design of the study, the primary aim was to verify whether there were differences in the efficacy of the various treatment schemes in pregnant infected COVID-19 patients with concomitant conditions at a high risk of disease progression. For high-risk disease progression, we mean hospitalization for COVID-19, progression to severe or critical respiratory requiring supplemental oxygen (severe illness), or mechanical ventilation (critical illness). The negativization of the infection was proven with the nasopharyngeal antigen test. As regards the effectiveness of the treatments, these included the total time (days) of negativization of the infection, the percentage of patients recovered in the absence of complications in the general state of health of the mother or the fetus, or both, and the percentage of patients with clinical worsening such as to require hospitalization. Possible adverse events related to the type of treatment undertaken were also evaluated and divided into infusion-related reactions (including hypersensitivity reactions) and molecule-dependent side effects.

### 2.5. Statistical Analysis

All statistical analyses were performed using SPSS software. Values are given as medians with ranges and means with standard deviations. Non-parametric Mann–Whitney tests were applied to compare the differences in values. All statistical analyses were considered significant with *p*-values < 0.05. The difference between the measured quantities had to have a *p* ≤  0.05 to be considered statistically significant.

## 3. Results

### 3.1. Patients

A total of 22 pregnant patients were included in the study, with a mean age of 32.38 (±5.55), a mean height of 166.72 ± 6.08, a weight of 70 ± 10.08, and a mean BMI of 25.20 ± 3.59. Six of them had not received the anti-COVID-19 vaccination, while four of them had an incomplete vaccination cycle, and the remaining twelve had undergone the three scheduled doses of the vaccine. Half were in the second trimester of pregnancy and half in the third. The first group included a pregnant woman with twin pregnancy, while the last group included a patient in active labor. Regarding symptoms, a total of 13 patients had fever, 11 patients reported pharyngodynia and rhinorrhea, 10 patients presented cough, 10 patients reported musculoskeletal pain, 9 patients reported headache, 4 patients had asthenia, and one of them had chest pain associated with respiratory distress. A total of 9 patients had two symptoms, 10 patients reported three symptoms, and the remaining 3 reported more than four symptoms. Among the main risk factors present at the time of observation, 15 had signs of immunosuppression (one patient was infected with HIV), 3 had a history of oncological pathology, 2 reported bronchial asthma, 1 had diabetes mellitus, 2 had pre-existing cardiovascular diseases at pregnancy, and finally, 1 was affected by ulcerative colitis. A total of 4 patients presented conditions referable to pregnancy, 1 had gestational diabetes mellitus, 1 had gestosis, 1 had idiopathic thrombocytopenia, and finally one of them reported a previous Toxoplasma gondii infection (Table 2).

### 3.2. Aim Analysis

From the point of view of the aim analysis, we examined the results related to the severity the COVID-19 infection, the impact of the infection on maternal comorbidities, the incidence of fetal and neonatal morbidity and mortality, and finally, we qualitatively and quantitatively evaluated the adverse events in the delivery methods. Comorbidities included pre-existing diabetes mellitus, hypertension, cardiovascular disease and HIV co-infection, and metabolic and autoimmune diseases. Of the 22 patients treated, none had a worsening of their infection or general state of health; 2 patients were hospitalized to perform treatment under medical supervision (one treated with Imdevimab/casirivimab and one treated with Bamlanivimab/etesevimab). A group of 5 patients was treated in the isolation rooms of the emergency department of our hospital (3 treated with Imdevimab/casirivimab, 1 with sotrovimab, and 1 with Bamlanivimab/etesevimab. The remaining 15 were treated in the outpatient setting (3 with Imdevimab/casirivimab and 12 with sotrovimab). No patients had grounds for further hospitalization for any other causes of illness. Two hospitalized patients presented immunodeficiency on admission and evidence of interstitial pneumonia not associated with severe respiratory insufficiency. Both patients had not adhered to the vaccination protocols against COVID-19. Of the 5 patients treated in the emergency department, 2 presented immunosuppression, and 3 other pregnant women had, respectively, comorbidity, bronchial asthma, diabetes mellitus, and oncological pathology in the course of follow-up. None of them had pneumonia or severe respiratory failure. Of these, 3 patients had not undergone complete vaccination while two had completed the entire vaccination cycle. The patients treated on an outpatient basis had undergone the entire vaccination cycle (Table 2). Up to six months of follow-up, we had no cases of death from any of the specified causes. None of the patients experienced a progression of outcomes related to COVID-19 infection. Among the treated patients, the decline in viral load up to negativity on the seventh day was observed in 5 (23%) patients, at two weeks it increased to 15 patients (68%), at three weeks 18 patients (82%), and the remaining 4 took more than three weeks to become negative (3 treated with Imdevimab/casirivimab and 1 treated with Bamlanivimab/etesevimab. Three of them were not in compliance with the vaccination schedule. The mean negativization time was 14.2 ± 7.63 days (median value 11.5), and in particular, 19.667 ± 2.91 days (median value 15) in the group treated with Imdevimab/casirivimab or Bamlanivimab/etesevimab and 9.07 ± 0.7 days (median value 10) in the group treated with sotrovimab. There were significative statistical differences between the negativization time of the sotrovimab group in comparison to Imdevimav/casirivimab and Bamlanivimab/etesevimab (*p* = 0.001) (Figure 2). The time of negativization of the COVID-19 infection was evaluated in terms of adherence or not to the vaccination plan. In general, the mean time of negativization for the vaccinated was 11.07 ± 6.33 days (median value 10) compared to that of the unvaccinated 16.77± 8.87 days (median value 13), which is not quite significant (*p* = 0.082) (Figure 3). In the group treated with sotrovimab, vaccinated patients had a negativization time of 8.8 ± 2.8 days (median value 8.5) while unvaccinated patients had 10 ± 3 days (median value 10), *p* = 0.57. Comparing the two treatment regimens according to vaccination coverage, it was observed that patients vaccinated and treated with sotrovimab (median value: 8.5 days) had a shorter negative time than patients treated with the combinations of Bamlanivimab/etesevimab or Imdevimav/casirivimab (median value: 15 days); for Mann–Whitney, this difference was statistically significant (*p* = 0.028) (Figure 4). Using the Mann–Whitney test, in unvaccinated pregnant patients, the time of negativization with sotrovimab (median value: 10) was quite significantly earlier in comparison to Bamlanivimab/etesevimab or Imdevimav/casirivimab (median value: 19.5; *p* = 0.068) (Figure 5). However, the parameter of days of positivity in the subgroup under analysis (unvaccinated patients) had a normal distribution, since the Shapiro–Wilk test was not significant (*p* = 0.068). This led to the use of a parametric test. Therefore, applying the student’s t-test to compare the mean days of positivity between the Bamlanivimab/etesevimab- or Imdevimav/casirivimab-treated group and the sotrovimab-treated group, the difference was statistically significant in favor of the latter group (t = 2503; *p* = 0.043). Using the Mann–Whitney test, we found no statistically significant differences regarding the number or severity of symptoms, dividing the patients by treatment groups (*p* = 0.56). Considering the groups of pregnant women with comorbidities and those without, the negative times were similar in both groups (13.92 ± 7.89 vs. 14.63 ± 8.26; *p* = 0.84).

### 3.3. Adverse Events

In the pregnant subjects of our study, adverse events related to treatment were monitored, considering in particular diarrhea, fever, nausea and vomiting, post-infusion tachycardia, increase in blood pressure values, skin rash, headache, mucositis, hypotension, dizziness, dysgeusia, inappetence, abdominal pain, asthenia, and pruritus. From a laboratory point of view, the markers of hepatic stasis and lysis were mainly monitored. During and after treatment, no side effects or adverse events were reported in our group. In none of the cases was there a worsening of the state of health of the pregnant women nor signs of fetal distress

### 3.4. Aim Analysis of Pregnancy

All 22 women involved in the study brought their pregnancies to term regularly in the absence of risk to their own health and that of their newborns. In our groups, there were no events of preterm deliveries, 1 woman had a twin birth, and 1 was in active labor. Most of the cases had signs of imminent delivery with premature rupture of the membranes (n = 13), six pregnant women came to the hospital for the appearance of regular abdominal pain, while in two cases the delivery was completed with a hospitalization program. Women who had comorbidities during pregnancy did not show a worsening of the underlying pathological conditions. Ten patients underwent vaginal delivery. Another six patients underwent intrapartum cesarean section due to difficulties in completing the delivery. An additional six patients underwent elective cesarean section, with no specific changes in the COVID-19 antiviral treatment group. Only one patient delivered with an active infection and active treatment, while the remaining 21 arrived at the birth already negative. A total of 23 newborns were born free from COVID-19 infection. In the observation group, there were no complications during the peripartum period, and there were no infants with a low birth weight or delayed attachment to the mother’s breast. The hospitalizations at the obstetrics and gynecology unit ended regularly within a maximum of 7 days.

## 4. Discussion

Pregnant women are at high risk of infectious viral diseases, such as COVID-19 infection, which are associated with physiological changes in the respiratory, circulatory, secretory, and immune systems during pregnancy [26]. There are several reasons to explain the possibility that COVID-19 colonizes the uteroplacental system; first of all, as already mentioned, the large abundance at the placental level, in the syncytiotrophoblast, in the cytotrophoblast, in the endothelium, in the vascular smooth muscle of the primary and secondary villi of ACE2 [27,28], as well as in conditions unrelated to pregnancy, is also expressed in the ovary, uterus, and vagina [29]. Furthermore, recent evidence related to the demonstration of the presence of SARS-CoV-2 viral RNA and protein in the placenta and the presence of virions found within the syncytiotrophoblast suggests that COVID-19 can infect the placenta [30,31,32]. Another study demonstrated [33] the presence of viral RNA in amniotic fluid and neonatal blood taken at birth. All this demonstrates that there is a potential risk of transmission of the COVID-19 infection from mother to fetus, and therefore why pregnant women have been included in the categories, being particularly at risk of serious complications.

Our study, although it was performed on a small number of pregnant patients, demonstrated the high safety and efficacy profile of three specific monoclonal antibody formulations for the treatment of COVID-19 infection in high-risk subjects of clinical worsening. Since their approval in the field of COVID-19 therapy, leading monoclonal antibody-based therapies such as the Casirivimab/imdevimab combination, the Bamlanivimab/etesevimab combination, and sotrovimab have demonstrated great efficacy in combating especially the delta and omicron variants in the general population. However, pregnant women in early trials of these drugs were excluded from the trial. Subsequently, when their use was also approved for pregnant women at a high risk of developing the infection, the evidence was scarce for the numerically limited series [34]. The first studies evaluating and reporting outcomes of pregnant women with COVID-19 and conducted with the combinations of Bamlanivimab/etesevimab and Casirivimab/imdevimab observed that monoclonal antibodies were well-tolerated and no adverse mother–fetus effects were reported [10,35,36]. Furthermore, another study similar to ours showed that adverse events after monoclonal antibody treatment were mild and rare [37]. However, neonatal outcomes have not been fully described due to a short follow-up period, while our study had a period of follow-up that was prolonged until 6 months. However, these data, plus our study, collectively suggest that COVID-19 monoclonal antibodies are well-tolerated and likely safe during pregnancy, such that the benefits of use may outweigh the potential risks. Already known evidence that human IgG1 (immunoglobulin G1) antibodies cross the placental barrier has contributed to the extension of the use of monoclonal antibodies during pregnancy, although it is not yet known whether the potential transfer of these drugs could represent an advantage or a risk to the developing fetus [38]. However, the presence of an action directly targeting the spike protein of the virus in the absence of other cross-reactivities seems to hold promise for avoiding adverse effects on the developing fetus. The efficacy of monoclonal antibodies is an interesting piece of information for cases of COVID-19 that arise during pregnancy given that fetus exposure to the infection most likely occurs after placenta maturation at the beginning of the second trimester, and also because monoclonal antibody therapies cross the placenta after this period [38]. This study, albeit with the limitations already mentioned, demonstrated that the use of the three formulations had a potential benefit, which justifies taking into consideration other associated health factors including the presence of comorbidities, cases of difficult pregnancy, or the concurrent presence of pregnancy-related disease. In our series, a pregnant woman in active labor was also treated, and the study demonstrated a certain safety of monoclonal antibodies, such as not affecting breastfeeding in the first few days. These data are in agreement with previous data that showed the absence of serious treatment-related adverse events [10,35,36,37]. The low frequency of maternal–fetal adverse events may be explained by the younger age of the patients with few comorbidities and the exchanged maternal IgG between the mother and the newborn during the first few days after birth.

Monoclonal antibodies bind to epitopes of the receptor binding domain (RBD) of the spike protein of the COVID-19 virus, thus preventing the interaction between the RBD and its human receptor ACE2 (angiotensin-converting enzyme 2) and consequently blocking the entry of the virus into cells [38]. ACE is an enzyme with a blood pressure regulatory function, present in our body in two different isoforms: ACE1 and ACE2 [39]. Some studies have shown that pregnancy is associated with an increase in the expression of ACE2, and therefore in this category of people, especially those with a smoking habit, the susceptibility to infection with COVID-19 is increased [40]. During pregnancy, ACE2 regulates the systemic arterial pressure of the pregnant woman and that of the maternal–fetal circuit, and this could favor the infection. The receptor ACE2 is also associated with a regulation of the immune response involving the release of cytokines in response to the replication of the viral genome [41]. Our study suggested that sotrovimab was the drug that achieved the greatest efficacy. Although there were no complications during pregnancy or in the prenatal period for all treatments, as demonstrated by other studies [42,43,44], sotrovimab obtained negativization of the infection in less time, especially in the group of pregnant women who were up to date with their vaccination schedule. Contrary to what was reported in a large meta-analysis, it does not depend on vaccination status [37]. In fact, in our study, we compared patients according to the vaccination course at the time of infection. The combinations were as effective as sotrovimab in eradicating the infection. However, only sotrovimab showed that it significantly induced rapid remission of the infection in both unvaccinated and vaccinated pregnant women, thereby conferring, as already known from the data in the literature, a longer coverage that could extend up to delivery and the first months of neonatal life. In particular, we found statistically significant differences for sotrovimab compared to the two combinations in vaccinated patients, but we still found a tendentially significant trend of obtaining more rapid negativization, even in non-vaccinated patients. This phenomenon could be explained by the fact that sotrovimab is particularly effective on the Omicron variant compared to the others and can provide greater protection, both in vaccinated patients with sera that are not very sensitive to this variant and in unvaccinated patients, extending coverage to this latter variant as well. In fact, although in this study it was not possible to identify the COVID-19 variant for each of the pregnant women, it is true that the observation period coincided with the maximum diffusion of the Omicron variant in Italy. The success of sotrovimab, as a drug capable of counteracting more variants and having a longer half-life than others, is extremely important if we think of the fact that the anti-COVID-19 vaccination campaigns have not been successful in all countries. Furthermore, vaccines may not work or be effective for all pregnant women [43]. Contrary to what has now been said, other studies argue that the most serious manifestations related to COVID-19 infection during pregnancy have been reported in women aged between 35 and 44 years and in the course of infections with the Alpha and Delta variants [10,44], and more recent data show that mild to moderate forms of infection such as those common to the Omicron variant do not have high numbers of adverse events [45,46]. From the data obtained, however, we believe that the results obtained from the use of sotrovimab are still interesting regardless of the severity of the clinical presentations, considering that in this category of patients, having shorter response times is certainly more important in terms of normal pregnancy management [47,48,49,50]. Most of the pregnant women enrolled in the study were immunocompromised and had a mix of other comorbidities, which could promote the evolution of the infection by taking advantage of an ineffective immune response to the infection or even the immunocompromise leads to the lack of antibody development in response to a complete vaccination course. From the data available in the literature, two studies have been published in which patients severely pathologically committed and with immunosuppression had a long-term persistence of the COVID-19 infection [51,52]. Immunocompromise is the most important cause of slowed virus shedding and therefore of prolongation of the disease, and although most immunocompromised individuals resolve the infection effectively, these cases highlight the potential risk of persistent infection with the development of viral variants resistant to current therapies [52].

For this reason, the evidence that sotrovimab worked in this class of patients by inducing shorter negative times represents preliminary data but is certain to be investigated by larger prospective studies, with the possibility of preventing the development of further variants precisely in the most fragile and immunocompromised patients, especially in particular conditions such as pregnancy. The main strength of this study was the congruous, long-term, longitudinal follow-up for neonatal and infant outcomes to detect neurodevelopmental disorders. The follow-up period is still ongoing and to date has not found adverse effects on maternal health or developmental complications in newborns. Since the specific information available was limited on the relationship between developmental disorders and the use of specific monoclonal antibodies for COVID-19 during pregnancy, we focused above all on the early emergence of neurodevelopmental disorders including autism spectrum disorder (ASD), attention-deficit/hyperactivity disorder (ADHD), intellectual disabilities, and specific learning disabilities, among others.

One of the limitations of the study was the size of the population, which could explain the limited value of maternal–fetal complications. Probably having a numerically larger series could have brought out a greater number of complications related to childbirth or neonatal complications. Although, in agreement with the data available in the literature, we consider that the use of monoclonal antibodies involves many more benefits than risks in the case of pregnancy. The lack of an adequate sample size limits the ability to determine the efficacy of treatments, although it has allowed some speculation to be made using non-parametric tests suitable for small sample sizes.

Another limitation could be linked to the fact that our data referred to the last two trimesters of pregnancy, while it cannot be excluded that treatment with monoclonal antibodies is dangerous in the first trimester of pregnancy, in which the fetus is more vulnerable to developmental complications, and therefore may have feto-maternal outcomes or adverse reactions related to these drugs.

A future direction, concerning the direct interaction between specific monoclonal antibodies for COVID-19 and developmental disorders, should include studies investigating the use of monoclonal antibodies against COVID-19 in the first trimester of pregnancy in order to observe their direct impact on neonatal development disorders and thus also certify the safety profile in this stage of gestation.

Other future directions on this issue could be greater knowledge on the optimal timing of monoclonal antibody administration during pregnancy to maximize efficacy and safety. This could lead to the emergence of new knowledge on prophylaxis in pregnant subjects at a high risk of exposure or in high-risk classes in general. From this point of view, the development of new monoclonal antibodies with a longer half-life and simpler administration could lengthen the period of protection against infection to prevent serious complications.

Another direction would be to broaden the study of knowledge on the topic through large-scale prospective and long-term follow-up studies, perhaps multicenter, in order to monitor the health and development of children born to mothers who received specific monoclonal antibodies for COVID-19 during pregnancy. This would help identify any potential long-term effects of the treatment. This would also allow the improvement of the safety profile as the evidence of the use of these antibodies in pregnant subjects becomes more numerous, as any specific adverse events that could be relevant for this population are not yet fully known.

It would also be useful to promote prospective large-scale comparative trials in order to compare the safety and efficacy of different COVID-19-specific monoclonal antibodies in pregnant subjects to determine which ones are the most suitable for use during pregnancy. This would allow for the development of specific guidelines and recommendations for the use of COVID-19-specific monoclonal antibodies in pregnant individuals based on evolving evidence.

In conclusion, this study clarifies that COVID-19 monoclonal antibodies are effective and safe during pregnancy, adding to the scarce evidence in the literature of a good safety profile and demonstrating that in particular, sotrovimab in pregnant women obtains better results than the comparison combinations by accelerating the negative times and providing longer-lasting coverage regarding the risk of reinfection due to its greater half-life.

## Figures and Tables

**Figure 1 microorganisms-11-01953-f001:**
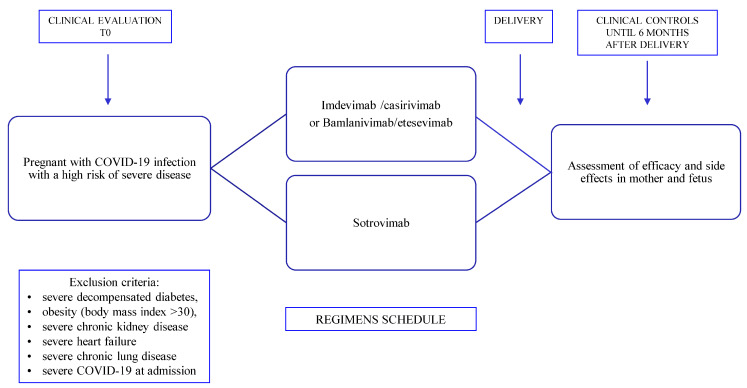
Design of the study.

**Figure 2 microorganisms-11-01953-f002:**
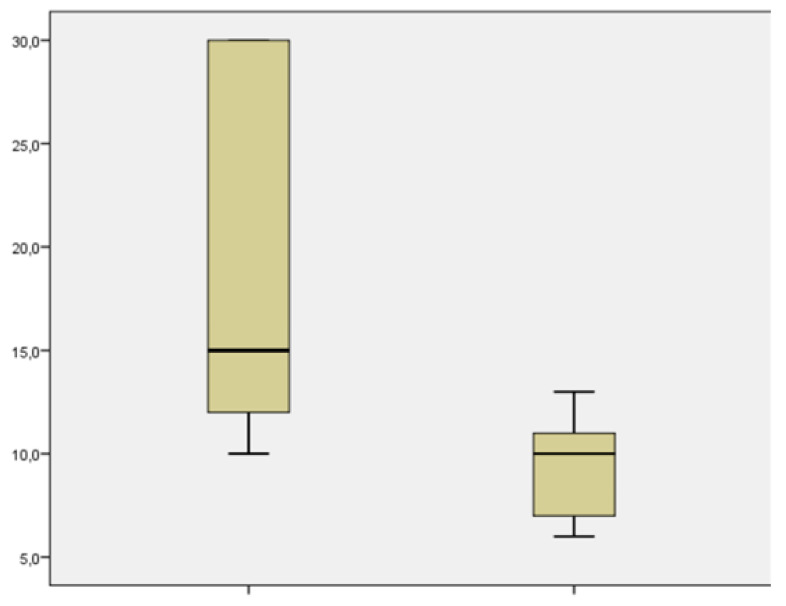
Patients’ characteristics and outcome regarding the two treatments. Time of negativization with sotrovimab is earlier in comparison to Imdevimav/casirivimab and Bamlanivimab/etesevimab. (*p* = 0.001).

**Figure 3 microorganisms-11-01953-f003:**
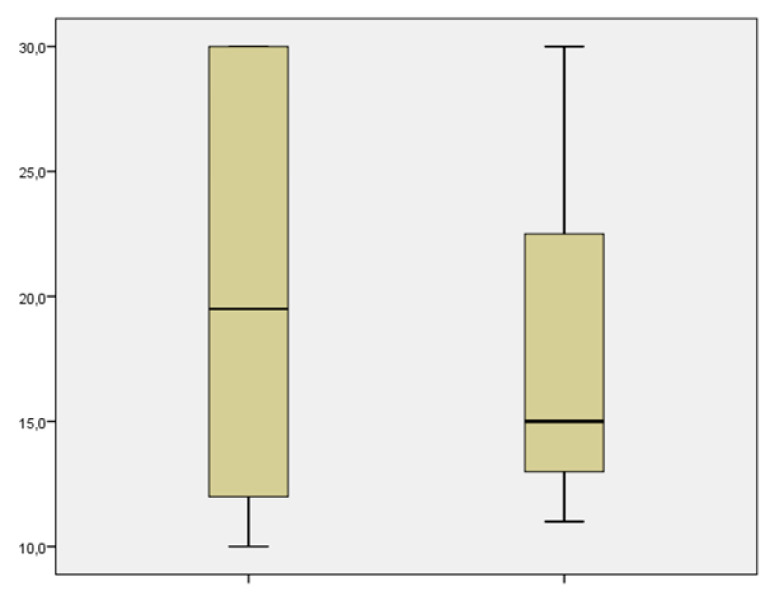
Patients’ characteristics and outcome regarding the two treatments. Time of negativization in vaccinated patients is not significantly earlier in comparison to no-vaccinated patients (*p* = 0.082).

**Figure 4 microorganisms-11-01953-f004:**
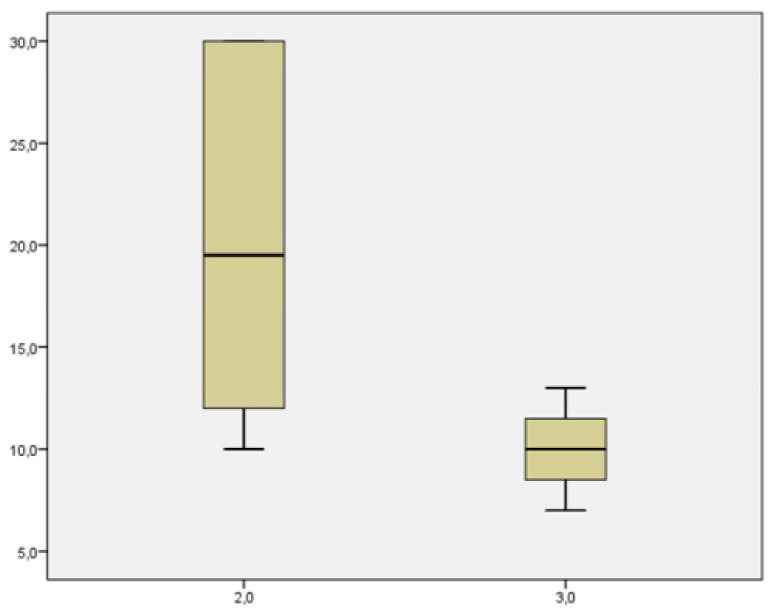
Time of negativization in vaccinated patients in the two treatment groups. The time of negativization in vaccinated patients treated with sotrovimab is quite significantly earlier in comparison to patientsreated with Bamlanivimab/etesevimab or lmdevimav/casirivimab (*p* = 0.068). Considering t di student the difference between the two groups is statistically significant (*p* = 0.043).

**Figure 5 microorganisms-11-01953-f005:**
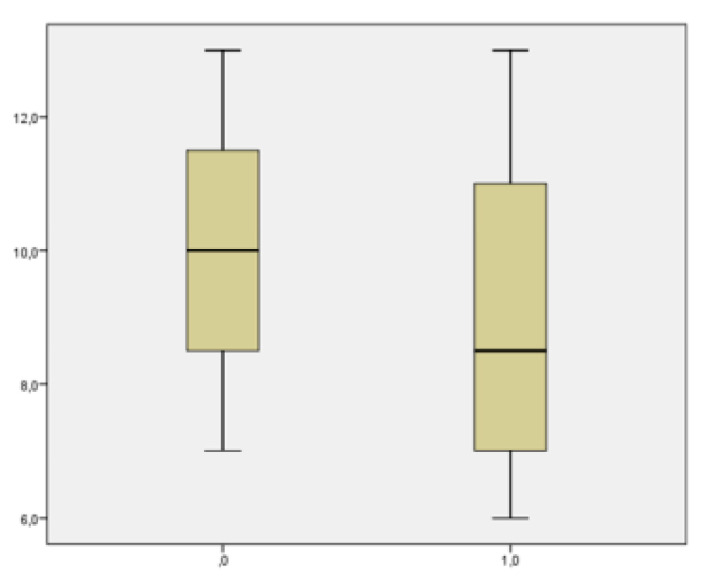
Patients’ characteristics, severity of symptoms and outcome. Time of negativization with sotrovimab is similar in comparison to Imdevimav/casirivimab and Bamlanivimab/etesevimab. (*p* < 0.0005).

**Table 1 microorganisms-11-01953-t001:** Risk factors indicating monoclonal antibody use in pregnant women.

Risk Factors
Body Mass Index (BMI) ≥ 30
Chronic renal failure, including peritoneal dialysis or hemodialysis
Uncontrolled diabetes mellitus (HbA lc > 9.0% 75 mmol/L) or with chronic complications
Primary or secondary immunodeficiency
Cardio-cerebrovascular disease (including hypertension with concomitant organ damage)
Chronic obstructive pulmonary disease and/or other chronic respiratory disease
Chronic liver disease
Pathologies of neurodevelopment and neurodegenerative pathologies
Hemoglobinopathies

**Table 2 microorganisms-11-01953-t002:** Patient’s characteristics at admission and outcome.

Patient Characteristics (22 pts)	
Age (mean ± SD)	32.38 ± 5.55
BMI (mean ± SD)	25.20 ± 3.59
Vaccination status	12 complete 6 incomplete 4 none
Pregnancy phase	12 second trimester 12 third trimester
General comorbidities	15 immunosuppressive status (one patient with HIV infection) 3 oncological pathology 2 bronchial asthma 1 diabetes mellitus 2 pre-existing cardiovascular diseases 1 ulcerative colitis
Pregnancy-related conditions (4 pts)	1 gestational diabetes mellitus 1 gestosis 1 idiopathic thrombocytopenia 1Toxoplasma gondii infection
Particular situation (2 pts)	1 twin pregnancy 1 active labor
Pregnancy outcome	12 Cesarian sections at COVID-19 (6 pts elective cesarean section) 10 ln term vaginal births 0 Precocious or Later birth
Time of negativization(mean ± SD)	14.2 ±7.63 days
COVID-19 status at birth	21 negative 1 positive

## Data Availability

Not applicable.

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
