# Peer review of "Use of Monoclonal Antibodies in Pregnant Women Infected by COVID-19: A Case Series"

_microorganisms, 2023, doi:10.3390/microorganisms11081953_

Round 1

Reviewer 1 Report

Dear Authors,

thank you for this speciffic article concerning Covid-19 pandemic. Please, use a proper template in the first place (the one you used is from 2020).

Here are some suggestions how to improve your paper:

1. Although the whole introduction is well prepared and nicely written, please, in the introducion add the interesting aspect on how the developmental pattern is connected to maternal and neonatal microbiome, see: - Nardi GM, Grassi R, Ndokaj A, Antonioni M, Jedlinski M, Rumi G, Grocholewicz K, Dus-Ilnicka I, Grassi FR, Ottolenghi L, Mazur M. Maternal and Neonatal Oral Microbiome Developmental Patterns and Correlated Factors: A Systematic Review-Does the Apple Fall Close to the Tree? Int J Environ Res Public Health. 2021 May 23;18(11):5569. doi: 10.3390/ijerph18115569.

2. Delete "including" from line 125 - replace it with "eg." instead, as this is exclusion criterion and might consufe the reader.

3. Please, add the weeks of pregnancy of the women.

Beside that small aspects, the article in general is well written and the discussion is well driven

Thank you.

Author Response

  1. Although the whole introduction is well prepared and nicely written, please, in the introduction add the interesting aspect on how the developmental pattern is connected to maternal and neonatal microbiome, see: - Nardi GM, Grassi R, Ndokaj A, Antonioni M, Jedlinski M, Rumi G, Grocholewicz K, Dus-Ilnicka I, Grassi FR, Ottolenghi L, Mazur M. Maternal and Neonatal Oral Microbiome Developmental Patterns and Correlated Factors: A Systematic Review-Does the Apple Fall Close to the Tree? Int J Environ Res Public Health. 2021 May 23;18(11):5569. doi: 10.3390/ijerph18115569.

R: according to the suggestion of the reviewer, a comment and the indicated reference has been included in the study.

  1. Delete "including" from line 125 - replace it with "eg." instead, as this is exclusion criterion and might confuse the reader.

R: according to the suggestion of the reviewer, the phrase has been corrected in the text.

  1. Please, add the weeks of pregnancy of the women.

R: according to the suggestion of the reviewer, the age of pregnancy has been added in the text.

Reviewer 2 Report

Introduction The introduction provides a general background on COVID-19 infection during pregnancy and the importance of studying the use of monoclonal antibodies in pregnant women. However, it lacks clarity and organization. The information presented is scattered and not well-structured, making it difficult for the reader to follow the main points. The introduction should be revised to provide a clear and concise overview of the topic, including the gap in knowledge regarding the use of monoclonal antibodies in pregnant women and the need for further research.

Patients and Methods???
The section describes the study design and patient inclusion criteria. However, it lacks important details that are necessary to evaluate the study's methodology. The information provided is insufficient to assess the representativeness of the study population and the reliability of the results. Additionally, there is no mention of ethical considerations, such as obtaining ethical approval and informed consent. The section should be expanded to provide more information on the study design, sample size calculation, patient recruitment process, ethical considerations, and any limitations of the study.

Results:
The results section presents the findings of the study, including patient characteristics, treatment outcomes, and adverse events. However, the results are reported in a narrative format without appropriate statistical analysis or presentation of data. The lack of statistical analysis limits the interpretation of the results and makes it difficult to assess the significance of the findings. The results should be presented using appropriate statistical measures, such as means, standard deviations, p-values, and confidence intervals, where applicable. Clear tables or figures should also be included to enhance the presentation of the data.

Discussion:
The discussion section provides a brief interpretation of the results but lacks a critical analysis of the findings in the context of existing literature. There is no discussion of the limitations of the study, potential biases, or alternative explanations for the results. The discussion should be expanded to include a thorough analysis of the strengths and weaknesses of the study, a comparison of the findings with previous research, and a discussion of the implications and significance of the results. Additionally, suggestions for future research directions could be included to further contribute to the field.

Overall, the manuscript has several shortcomings in each section. The abstract, introduction, results, and discussion sections need to be revised and improved to enhance clarity, organization, statistical analysis, critical analysis of the findings, and the inclusion of relevant information.

Why no figure found in the file???

The English and grammar in this manuscript have several issues. Here are some of the main problems:

  1. Lack of Clarity: The abstract and introduction sections contain numerous run-on sentences, making the text difficult to follow and understand. There is a need for clearer sentence structure and organization. Example: "Bamlanivimab and etesevimab are a monoclonal antibody combination for the treatment of COVID-19 that have the same mechanism of action as the combination of imdevimab and casirivimab. The sentence is structurally unclear and confusing due to its long and convoluted structure.

  2. Wordiness: The manuscript often uses excessive and convoluted language, resulting in wordy and unclear sentences. Simplifying the language and improving sentence structure would enhance readability.
    Example: "Taking into account that over 5 million people worldwide have died from COVID-19 infection due to its most dramatic manifestations such as respiratory failure and acute respiratory distress syndrome (19), it was especially necessary to protect the main categories at risk, subjecting them to a widespread vaccination campaign (20). This sentence could be simplified by removing unnecessary phrases and dividing it into shorter, clearer sentences.

  3. Inconsistent Tense: The manuscript switches between past and present tense inconsistently. It should maintain a consistent tense throughout the text for clarity and coherence.
    Example: "The patients were followed up one week, one month and 6 months after delivery." The use of past tense ("were followed") and present tense ("after delivery") in the same sentence creates inconsistency.

  4. Poor Sentence Structure: Many sentences are structurally incorrect, making them difficult to comprehend. Correcting sentence structure errors would improve readability.
    Example: "Of the 22 patients treated with the three different therapeutic protocols, none had a worsening of the infection and of the general state of health, although as a precaution 2 patients were hospitalized and performed the treatment under medical supervision." This sentence is long and contains multiple ideas, making it difficult to follow. It would benefit from being divided into shorter, more concise sentences.

  5. Lack of Proper Punctuation: The manuscript lacks proper punctuation in various instances. Missing commas, incorrect placement of periods, and absence of quotation marks contribute to confusion and ambiguity.
    Example: "All statistical analyses were considered significant with p -values less than < 0.05." The incorrect placement of the less-than symbol ("<") and the lack of a space between "p" and "-values" contribute to the confusion.

  6. Typographical Errors: The manuscript contains several typographical errors, including misspellings, missing letters, and incorrect word usage. Proofreading for such errors is necessary.
    Example: "Patients included in this observational study were older than 18 years and tested positive for SARS-CoV-2 by reverse transcriptase-polymerase chain reaction (RT-PCR) test or an antigen test with a period of symptom onset not exceeding the previous 5 days." The phrase "with a period of symptom onset not exceeding the previous 5 days" is unclear and could be revised for clarity.

It is important to thoroughly edit and proofread the manuscript to address these language issues and improve its clarity and coherence.

Author Response

Introduction The introduction provides general background on COVID-19 infection during pregnancy and the importance of studying the use of monoclonal antibodies in pregnant women. However, it lacks clarity and organization. The information presented is scattered and not well-structured, making it difficult for the reader to follow the main points. The introduction should be revised to provide a clear and concise overview of the topic, including the gap in knowledge regarding the use of monoclonal antibodies in pregnant women and the need for further research.

R: according to the suggestion of the reviewer, the introduction has been revised in the text.

Patients and Methods???

The section describes the study design and patient inclusion criteria. However, it lacks important details that are necessary to evaluate the study's methodology. The information provided is insufficient to assess the representativeness of the study population and the reliability of the results. Additionally, there is no mention of ethical considerations, such as obtaining ethical approval and informed consent. The section should be expanded to provide more information on the study design, sample size calculation, patient recruitment process, ethical considerations, and any limitations of the study.

R: In accordance with the reviewer's suggestions, the study design was better explained also with the help of a figure (Figure 1); the ethical opinion on the study was made explicit and the informed consent to the study was signed by one of the pregnant women representing the whole group of patients included. With regard to the calculation of the sample size, it must be said that we are presenting a case series, from which we have deduced some data that we have postulated for statistical calculation considering the reduced number of cases and with an essential fact that we have immediately clarified, namely that both treatment schemes were safe for pregnant women and for the fetus, underlining only the superiority of sotrovimab in obtaining less time to eradicate the viral infection, even in unvaccinated patients. It is clear that this finding is the result of the review of a case series and therefore we postpone the confirmation or denial of this data to future more extensive trials or subsequent meta-analyses. Moreover, in a case series study, the calculation of sample size is not always a strict requirement. Case series studies are observational studies that describe the characteristics or outcomes of a group of individuals who have a particular condition or exposure. These studies typically involve a small number of cases and are often used to generate hypotheses or explore rare conditions. Unlike analytical studies (for example, randomized controlled trials, cohort studies, or case-control studies), case series studies typically do not aim to test a specific hypothesis or establish causal relationships. Instead, they focus on providing descriptive information about a specific group of cases. Because the use of hypothesis testing or statistical comparisons is optional in case series studies, sample size calculations are not a standard part of the study design. Researchers conducting case series studies generally aim to include all available cases during the study period rather than selecting a predetermined sample size based on statistical considerations.

Results:

The results section presents the findings of the study, including patient characteristics, treatment outcomes, and adverse events. However, the results are reported in a narrative format without appropriate statistical analysis or presentation of data. The lack of statistical analysis limits the interpretation of the results and makes it difficult to assess the significance of the findings. The results should be presented using appropriate statistical measures, such as means, standard deviations, p-values, and confidence intervals, where applicable. Clear tables or figures should also be included to enhance the presentation of the data.

R: In accordance with the reviewer's suggestions, it should be noted that the figures have been included as supplementary files as per editorial regulations and as the publisher himself can confirm, complete with statistical representation and calculation. In the text, there is also a reference to figures and tables, which however have been sent separately. The reviewer probably only downloaded the text.

Discussion:

The discussion section provides a brief interpretation of the results but lacks a critical analysis of the findings in the context of existing literature. There is no discussion of the limitations of the study, potential biases, or alternative explanations for the results. The discussion should be expanded to include a thorough analysis of the strengths and weaknesses of the study, a comparison of the findings with previous research, and a discussion of the implications and significance of the results. Additionally, suggestions for future research directions could be included to further contribute to the field.

R: In accordance with the reviewer's suggestions, the discussion was revised and enriched with the missing parts

Overall, the manuscript has several shortcomings in each section. The abstract, introduction, results, and discussion sections need to be revised and improved to enhance clarity, organization, statistical analysis, critical analysis of the findings, and the inclusion of relevant information.

Why no figure found in the file???

R: In accordance with the reviewer's suggestions, it should be noted that the figures have been adapted to the comments expressed during the peer review and that in any case they had already been included as supplementary files as per editorial standards and as the publisher himself can confirm. In the text, there is also a reference to figures and tables, which however have been sent separately.

Comments on the Quality of English Language

The English and grammar in this manuscript have several issues. Here are some of the main problems:

Lack of Clarity: The abstract and introduction sections contain numerous run-on sentences, making the text difficult to follow and understand. There is a need for clearer sentence structure and organization. Example: "Bamlanivimab and etesevimab are a monoclonal antibody combination for the treatment of COVID-19 that have the same mechanism of action as the combination of imdevimab and casirivimab. The sentence is structurally unclear and confusing due to its long and convoluted structure.

R: according to the suggestion of the reviewer, the phrase has been corrected in the text.

Wordiness: The manuscript often uses excessive and convoluted language, resulting in wordy and unclear sentences. Simplifying the language and improving sentence structure would enhance readability.

Example: "Taking into account that over 5 million people worldwide have died from COVID-19 infection due to its most dramatic manifestations such as respiratory failure and acute respiratory distress syndrome (19), it was especially necessary to protect the main categories at risk, subjecting them to a widespread vaccination campaign (20). This sentence could be simplified by removing unnecessary phrases and dividing it into shorter, clearer sentences.

R: according to the suggestion of the reviewer, the phrase has been corrected in the text.

Inconsistent Tense: The manuscript switches between past and present tense inconsistently. It should maintain a consistent tense throughout the text for clarity and coherence.

Example: "The patients were followed up one week, one month and 6 months after delivery." The use of past tense ("were followed") and present tense ("after delivery") in the same sentence creates inconsistency.

R: according to the suggestion of the reviewer, the phrase has been corrected in the text.

Poor Sentence Structure: Many sentences are structurally incorrect, making them difficult to comprehend. Correcting sentence structure errors would improve readability.

Example: "Of the 22 patients treated with the three different therapeutic protocols, none had a worsening of the infection and of the general state of health, although as a precaution 2 patients were hospitalized and performed the treatment under medical supervision." This sentence is long and contains multiple ideas, making it difficult to follow. It would benefit from being divided into shorter, more concise sentences.

R: according to the suggestion of the reviewer, the phrase has been corrected in the text.

Lack of Proper Punctuation: The manuscript lacks proper punctuation in various instances. Missing commas, incorrect placement of periods, and absence of quotation marks contribute to confusion and ambiguity.

Example: "All statistical analyses were considered significant with p -values less than < 0.05." The incorrect placement of the less-than symbol ("<") and the lack of a space between "p" and "-values" contribute to the confusion.

R: according to the suggestion of the reviewer, the phrase has been corrected in the text.

Typographical Errors: The manuscript contains several typographical errors, including misspellings, missing letters, and incorrect word usage. Proofreading for such errors is necessary.

Example: "Patients included in this observational study were older than 18 years and tested positive for SARS-CoV-2 by reverse transcriptase-polymerase chain reaction (RT-PCR) test or an antigen test with a period of symptom onset not exceeding the previous 5 days." The phrase "with a period of symptom onset not exceeding the previous 5 days" is unclear and could be revised for clarity.

R: according to the suggestion of the reviewer, the phrase has been corrected in the text.

It is important to thoroughly edit and proofread the manuscript to address these language issues and improve its clarity and coherence.

R: According to the suggestions of the reviewer in English it will be revised with the help of the editor.

Round 2

Reviewer 2 Report

 the manuscript has several shortcomings in each section.

The English and grammar in this manuscript have several issues.